# Deep Policies for Online Bipartite Matching: A Reinforcement Learning Approach

**Mohammad Ali Alomrani**                    *mohammad.alomrani@mail.utoronto.ca*
*Department of Electrical & Computer Engineering*
*University of Toronto*

**Reza Moravej**                    *mreza.moravej@mail.utoronto.ca*
*Department of Mechanical & Industrial Engineering*
*University of Toronto*

**Elias B. Khalil**                    *khalil@mie.utoronto.ca*
*Department of Mechanical & Industrial Engineering*
*SCALE AI Research Chair in Data-Driven Algorithms for Modern Supply Chains*
*University of Toronto*

**Reviewed on OpenReview:** *https://openreview.net/forum?id=mbwm7NdkpO*

## Abstract

The challenge in the widely applicable online matching problem lies in making irrevocable assignments while there is uncertainty about future inputs. Most theoretically-grounded policies are myopic or greedy in nature. In real-world applications where the matching process is repeated on a regular basis, the underlying data distribution can be leveraged for better decision-making. We present an end-to-end Reinforcement Learning framework for deriving better matching policies based on trial-and-error on historical data. We devise a set of neural network architectures, design feature representations, and empirically evaluate them across two online matching problems: Edge-Weighted Online Bipartite Matching and Online Submodular Bipartite Matching. We show that most of the learning approaches perform consistently better than classical baseline algorithms on four synthetic and real-world datasets. On average, our proposed models improve the matching quality by 3–10% on a variety of synthetic and real-world datasets.Our code is publicly available at `https://github.com/lyeskhalil/CORL`.

## 1 Introduction

Originally introduced by Karp et al. (1990), the Online Bipartite Matching (OBM) problem is a simple formulation of sequential resource allocation. A fixed set $U$ of known entities (e.g., ads, tasks, servers) are to be dynamically assigned to at most one of a discrete stream $V$ of (apriori unknown) entities (e.g., adslots, job candidates, computing job) upon their arrival, so as to maximize the size of the final matching. Matching decisions are irrevocable and not matching is allowed at all times. Despite its simplicity, finding better algorithms for OBM and its variants remains an active area of research. The uncertainty about future inputs makes online problems inherently challenging. While practical exact methods (e.g., using integer programming formulations and solvers) exist for many offline combinatorial problems, the restriction to irrevocable and instant decision-making makes the use of such algorithmic tools impractical. Existing algorithms for online matching are thus typically myopic and greedy in nature (Mehta, 2013).

In practice, however, the underlying (bipartite) graph instances may come from the same unknown distribution (Borodin et al., 2020). In many applications, a sufficiently large collection of samples from the data can represent the often implicit statistical properties of the entire underlying data-generating distribution.

It is often the case that corporations, for example, have access to a wealth of information that is represented as a large graph instance capturing customer behaviour, job arrivals, etc. Thus, it is sensible for an algorithm to use historical data to derive statistical information about online inputs in order to perform better on future instances. However, the majority of non-myopic hand-designed algorithms depend on estimating the arrival distribution of the incoming nodes (Borodin et al., 2020; Mehta, 2013). The downside of this approach is that imperative information such as the graph sparsity, ratio of incoming nodes to fixed nodes, existence of community structures, degree distribution, and the occurrence of structural motifs are ignored. Ideally, a matching policy should be able to leverage such information to refine its decisions based on the observed history.

In this work, we formulate online matching as a Markov Decision Process (MDP) for which a neural network is trained using Reinforcement Learning (RL) on past graph instances to make near-optimal matchings on unseen test instances. We design six unique models, engineer a set of generic features, and test their performance on two variations of OBM across two synthetic datasets and two real-world datasets. Our contributions can be summarized as follows:

**Automating matching policy design:** Motivated by practical applications, other variants of OBM have been introduced with additional constraints (such as fairness constraints) or more complex objective functions than just matching size. Our method reduces the reliance on human handcrafting of algorithms for each individual variant of OBM since the RL framework presented herein can flexibly model them; this will be demonstrated for the novel Online Submodular Bipartite Matching problem (Dickerson et al., 2019).

**Deriving tailored policies using past graph instances:** We show that our method is capable of taking advantage of past instances to learn a near-optimal policy that is tailored to the problem instance. Unlike "pen-and-paper" algorithms, our use of historical information is not limited to estimating the arrival distribution of incoming nodes. Rather, our method takes advantage of additional statistics such as the existing (partial) matching, graph sparsity, the $|U|$-to-$|V|$ ratio, and the graph structure. Taking in a more comprehensive set of statistics into account allows for fine-grained decision-making. For example, the RL agent can learn to skip matching a node strategically based on the observed statistical properties of the current graph. Our results on synthetic and real world datasets demonstrate this.

**Leveraging Node Attributes:** In many variants of OBM, nodes have identities, e.g., the nodes in $V$ could correspond to social media users whose demographic information could be used to understand their preferences. Existing algorithms are limited in considering such node features that could be leveraged to obtain better solutions. For instance, the RL agent may learn that connecting a node $v$ with a particular set of attributes to a specific node in $U$ would yield high returns. The proposed framework can naturally account for such attributes, going beyond simple greedy-like policies. We will show that accounting for node attributes yields improved results on a real-world dataset for Online Submodular Bipartite Matching.

## 2 Problem Setting

In a bipartite graph $G = (U, V, E)$, $U$ and $V$ are disjoint sets of nodes and $E$ is the set of edges connecting a node in $U$ to one in $V$. In the online bipartite matching problem, the vertex set $U$ is fixed and at each timestep a new node $v \in V$ and its edges $\{(u, v) : u \in U\}$ arrive. The algorithm must make an *instantaneous* and *irrevocable* decision to match $v$ to one of its neighbors or not match at all. Nodes in $U$ can be matched to at most one node in $V$. The time horizon $T = |V|$ is finite and assumed to be known in advance. The simplest generalization of OBM is the edge-weighted OBM (E-OBM), where a non-negative weight is associated with each edge. Other well-known variants include Adwords, Display Ads and Online Submodular Welfare Maximization (Mehta, 2013). We will focus our experiments on E-OBM and Online Submodular Bipartite Matching (OSBM), a new variation of the problem introduced by Dickerson et al. (2019); together, the two problems span a wide range in problem complexity. The general framework can be extended with little effort to address other online matching problems with different constraints and objectives; see Appendix F for a discussion and results on Adwords.

## 2.1 Edge-weighted OBM (E-OBM)

Each edge $e \in E$ has a predefined weight $w_e \in \mathbb{R}^+$ and the objective is to select a subset $S$ of the incoming edges that maximizes $\sum_{e \in S} w_e$. Note that in the offline setting, where the whole graph is available, this problem can be solved in polynomial time using existing algorithms (Kuhn, 1955). However, the online setting involves reasoning under uncertainty, making the design of optimal online algorithms non-trivial.

## 2.2 Online Submodular Bipartite Matching (OSBM)

We first define some relevant concepts:

*Submodular function*: A function $f : 2^U \to \mathbb{R}^+$, $f(\emptyset) = 0$ is *submodular* iff $\forall S, T \subseteq U$:

$$f(S \cup T) + f(S \cap T) \leq f(S) + f(T).$$

Some common examples of submodular functions include the coverage function, piecewise linear functions, and budget-additive functions. In our experiments, we will focus on the weighted coverage function following Dickerson et al. (2019):

*Coverage function*: Given a universe of elements $U$ and a collection of $g$ subsets $A_1, A_2, \ldots, A_g \subseteq U$, the function $f(M) = | \cup_{i \in M} A_i|$ is called the coverage function for $M \subseteq \{1, \ldots, g\}$. Given a non-negative, monotone weight function $w : 2^U \to \mathbb{R}^+$, the weighted coverage function is defined analogously as $f(M) = w(\cup_{i \in M} A_i)$ and is known to be submodular.

In this setting, each edge $e \in E$ incident to arriving node $v_t$ has the weight $f(M_t \cup \{e\}) - f(M_t)$ where $M_t$ is the matching at timestep $t$. The objective in OSBM is to find $M$ such that $f(M) = \sum_{e \in M} w_e$ is maximized; $f$ is a submodular function.

An illustrative application of the OSBM problem, identified by Dickerson et al. (2019), can be found in movie recommendation systems. There, the goal is to match incoming users to a set of movies that are both relevant and diverse (genre-wise). A user can login to the platform multiple times and may be recommended (matched to) a movie or left unmatched. Since we have historical information on each user's average ratings for each genre, we can quantify diversity as the weighted coverage function over the set of genres that were matched to the user. The goal is to maximize the sum of the weighted coverage functions for all users. More concretely, if we let $U$ be the universe of genres, then any movie $i$ belongs to a subset of genres $A_i$. Let $L$ be the set of all users, $M_l$ be the set of movies matched to user $l$, and $f_l(M_l) = w(\cup_{i \in M} A_i)$ be the weighted coverage function defined as the sum of the weights of all genres matched to the user, where the weight of a genre $k$ is the average rating given by user $l$ to movies of genre $k$. Each user's weighted coverage function is submodular. The objective of OSBM is to maximize the (submodular) sum of these user functions: $f(M) = \sum_{l \in L} f_l(M_l)$.

## 2.3 Arrival Order

Online problems are studied under different input models that allow the algorithm to access varying amounts of information about the arrival distribution of the vertices in $V$. The *adversarial* order setting is often used to study the worst-case performance of an algorithm, positing that an imaginary adversary can generate the worst possible graph and input order to make the algorithm perform poorly. More optimistic is the *known i.i.d distribution (KIID)* setting, where the algorithm knows $U$ as well as a distribution $\mathcal{D}$ on the possible *types* of vertices in $V$. Each arriving vertex $v$ belongs to one type and vertices of a given type have the same neighbours in $U$. This assumption, i.e., that the arrival distribution $\mathcal{D}$ is given, is too optimistic for complex real-world applications.

In this work, we study the ***unknown i.i.d distribution (UIID)*** setting, which lies between the *adversarial* and the *KIID* settings in terms of how much information is given about the arrival distribution (Karande et al., 2011). The *unknown i.i.d setting* best captures real-world applications, where a base graph is provided from an existing data set, but an explicit arrival distribution $\mathcal{D}$ is not accessible. For example, a database of past job-to-candidate or item-to-customer relationships can represent a base graph.

It is thus safe to assume that the arriving graph will follow the same distribution. The arrival distribution is accessible on through sample instances drawn from $\mathcal{D}$. More details on data generation are provided in Section 5.1 and Appendix C.

## 3 Related Work

**Traditional Algorithms for OBM:** Generally, the focus of algorithm design for OBM has been on worst-case approximation guarantees for "pen-and-paper" algorithms via competitive analysis, rather than average-case performance in a real-world application. We refer the reader to (Karande et al., 2011; Mehta, 2013) for a summary of the many results for OBM under various arrival models. On the empirical side, an extensive set of experiments by Borodin et al. (2020) showed that the naive greedy algorithm performs similarly to more sophisticated online algorithms on synthetic and real-world graphs in the KIID setting. Though the experiments were limited to OBM with $|U| = |V|$, they were conclusive in that ($i$) greedy is a strong baseline in practical domains, and ($ii$) having better proven lower bounds does not necessarily translate into better performance in practice.

The main challenge in online problems is decision-making in the face of uncertainty. Many traditional algorithms under the KIID setting aim to overcome this challenge by explicitly approximating the distribution over node types via a type graph. The algorithms observe past instances and estimate the frequency of certain types of online nodes, i.e., for each type $i$, the algorithm predicts a probability $p_i$ of a node of this type arriving. We refer the reader to Borodin et al. (2020) for detailed explanation on algorithms under the KIID setting. As noted earlier, the KIID setting is rather simplistic compared to the more realistic UIID setting that we tackle here. Other non-myopic algorithms have been proposed which do not rely on estimating the arrival distribution. For example, Awasthi & Sandholm (2009) solve stochastic kidney exchange, a generalization of OBM, by sampling a subset of future trajectories, solving the offline problem on each of them, and assigning a score to each action. The algorithm then selects the action that is the best overall. To the best of our knowledge, our presented method is the first which learns a custom policy from data based on both explicit and implicit patterns (such as graph sparsity, the graph structure, degree distribution, etc.).

**Learning in Combinatorial Optimization:** There has recently been substantial progress in using RL for finding better heuristics for offline, NP-complete graph problems. Dai et al. (2017) presented an RL-based approach combined with graph embeddings to learn greedy heuristics for some graph problems. Barrett et al. (2020) take a similar approach but start with a non-empty solution set and allow the policy to explore by removing nodes/edges from the solution. Chen & Tian (2019) learn a secondary policy to pick a particular region of the current solution to modify and incrementally improve the solution. The work by Kool et al. (2019) uses an attention based encoder-decoder approach to find high-quality solutions for TSP and other routing problems. We refer to the following surveys for a more comprehensive view of the state of this area (Mazyavkina et al., 2021; Bengio et al., 2021).

Prior work on using predictive approaches for online problems has been fairly limited. Wang et al. (2019) overlook the real-time decision making condition and use Q-learning for a batch of the arriving nodes. The matching process, however, is not done by an RL agent but using an offline matching algorithm. Consequently, their method is not practical for OBM variants that are NP-hard in the offline setting (e.g., Adwords) and where instantaneous decision making is paramount, e.g., in ad placement on search engines. The work by Kong et al. (2019) is one of few to apply RL to online combinatorial optimization. Their work differs from ours in three main ways: ($i$) the question raised there is whether RL can discover algorithms which perform best on worst-case inputs. They show that the RL agent will eventually learn a policy which follows the "pen-and-paper" algorithm with the best worst-case guarantee. Our work, on the other hand, asks if RL can outperform hand-crafted algorithms on average; ($ii$) The MDP formulation introduced in (Kong et al., 2019), unlike ours, does not consider the entire past (nodes that have previously arrived and the existing matching) which can help an RL policy better reason about the future; ($iii$) our family of invariant neural network architectures apply to graphs of arbitrary sizes $|V|$ and $|U|$. More details about the advantages of our method are provided in the next section. Zuzic et al. (2020) propose a GAN-like adversarial training approach to learn robust OBM algorithms. However, just like (Kong et al.,

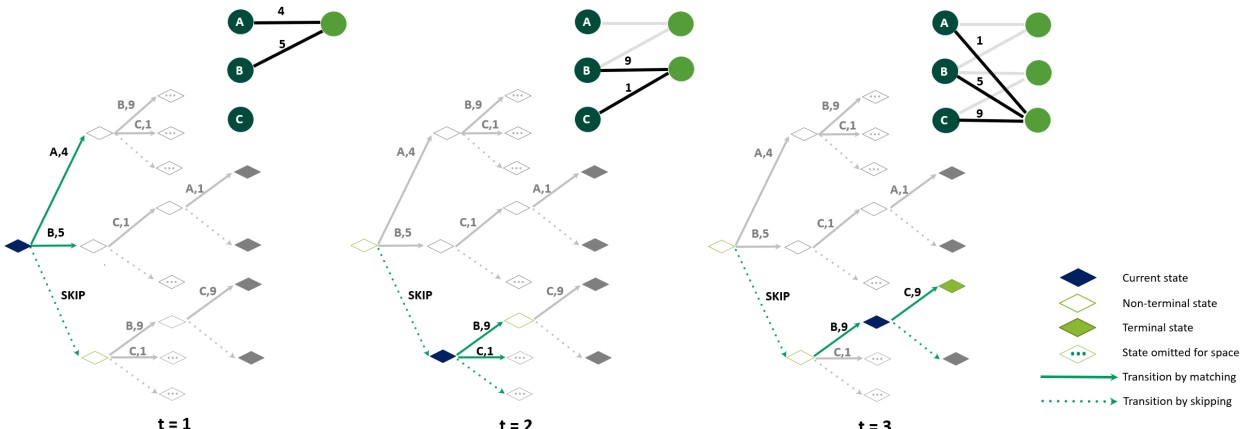

Figure 1: The MDP formulation of E-OBM. The agent is trained on different graph instances sampled from a distribution $\mathcal{D}$. At each timestep $t$, the agent picks a node to match or skip. In this 3x3 graph, the optimal matching has weight 22; following the greedy policy would yield a matching of weight 7. The illustrated policy ends up with a matching of weight 18.

2019), they are more concerned with learning algorithms that are robust to hard distributions rather than real-world graphs, and do not utilize historical information accumulated in the previous steps within the same instance.

**Online Algorithm Design via Learned Advice:** A hybrid paradigm has been recently introduced where the predictive and competitive-analysis approaches are combined to tackle online problems. Such algorithms take advantage of the predictions made by a model to obtain an improved competitive ratio while still guaranteeing a worst-case bound when the predictions are inaccurate. Work in this area has resulted in improvements over traditional algorithms for the secretary, ski rental and online matching problems (Antoniadis et al., 2020; Wang et al., 2020; Diakonikolas et al., 2021; Purohit et al., 2018). Unlike our approach, the model does not get to construct a solution. Rather, its output is used as advice to a secondary algorithm. Since competitive analysis is of concern in this line of work, the algorithm is not treated as a black box and must be explicitly handcrafted for each different online problem. On the other hand, we introduce a general end-to-end framework that can handle online matching problems with different objectives and constraints, albeit without theoretical performance guarantees.

## 4 Learning Deep Policies for Matching

We now formalize online matching as a Markov Decision Process (MDP). We then present a set of neural network architectures with different representational capabilities, numbers of parameters, and assumptions on the size of the graphs. An extensive set of features have been designed to facilitate the learning of high-performance policies. We conclude this section by mentioning the RL training algorithm we use as well as a supervised behavioral cloning baseline.

### 4.1 MDP Formulation

The online bipartite matching problem can be formulated in the RL setting as a Markov Decision Process as follows; see Fig. 1 for a high-level illustration. Each instance of the online matching problem is drawn uniformly at random from an unknown distribution $\mathcal{D}$. The following MDP captures the sequential decision-making task at hand:

– **State**: A state $S$ is a set of selected edges (a matching) and the current (partial) bipartite graph $G$. A terminal state $\hat{S}$ is reached when the final node in $V$ arrives. The length of an episode is $T = |V|$.

– **Action**: The agent has to pick an edge to match or skip. At each timestep $t$, a node $v_t$ arrives with its edges. The agent can choose to match $v_t$ to one of its neighbors in $U$ or leave it unmatched. Therefore, $|A_t|$, the maximum number of possible actions at time $t$ is $|\text{Ngbr}(v)| + 1$, where $\text{Ngbr}(v)$ is the set of $U$ nodes with edges to $v$. Note that there can exist problem-specific constraints on the action space, e.g., a fixed node can only be matched once in E-OBM. Unlike the majority of domains where RL is applied, the uncertainty is exogenous here. Thus, the transition is *deterministic* regardless of the action picked. That is, the (random) arrival of the next node is independent of the previous actions.

– **Reward function**: The reward $r(s, a)$ is defined as the weight of the edge selected with action $a$. Hence, the cumulative reward $R$ at the terminal state $\hat{S}$ represents the total weight of the final matching solution:

$$R = \sum_{e \in \hat{S}} w_e.$$

– **Policy**: A solution (matching) is a subset of the edges in $E$, $\pi = \bar{E} \subset E$. A stochastic policy, parameterized by $\theta$, outputs a solution $\pi$ with probability

$$p_\theta(\pi|G) = \prod_{t=1}^{|V|} p_\theta(\pi_t|s_t),$$

where $s_t$ represents the state at timestep $t$, $G$ represents the full graph, and $\pi_t$ represents the action picked at timestep $t$ in solution $\pi$.

## 4.2 Deep Learning Architectures

In this section, we propose a number of architectures that can be utilized to learn effective matching policies. Unless otherwise stated, the models are trained using RL.

**Feed-Forward** (`ff`): When node $v_t$ arrives, the `ff` policy will take as input a vector $(w_0, \ldots, w_{|U|}, m_0, \ldots, m_{|U|}) \in \mathbb{R}^{2(|U|+1)}$ [1], where $w_u$ is the weight of the edge from $v_t$ to fixed node $u$ (with $w_u = 0$ if $v$ is not a neighbor of $u$), and $m_u$ is a binary mask representing the availability of node $u$ for matching. The policy will output a vector of probabilities of size $|U| + 1$, where the additional action represents skipping. `ff` is similar to the architecture presented in Kong et al. (2019).

**Feed-Forward with history** (`ff-hist`): This model is similar to `ff` but takes additional historical information about the current graph to better reason about future input. That is, `ff-hist` will take in a vector consisting of five concatenated vectors, $(w, m, h_t, g_t, n_t)$. The vectors $w$ and $m$ are the same as those in `ff`. The feature vectors $h, n, g$ contain a range of node-level features such as average weights seen so far per fixed node and solution-level features such as maximum weight in current solution; see Table 1 for details.

**Invariant Feed-Forward** (`inv-ff`): We present an invariant architecture, inspired by Andrychowicz et al. (2016), which processes each of the edges and their fixed nodes *independently* using the same (shared) feed-forward network; see Fig. 2 for an illustration. That is, `inv-ff` will take as input a 3-dimensional vector, $(w_u, s_u, w_{mean})$, where $w_{mean}$ is the mean of the edge weights incident to incoming node $v_t$, and $s_u$ is a binary flag set to 1 if $u$ is the "skip" node. The output for each potential edge is a single number $o_u$. The vector $o$ is normalized using the softmax to output a vector of probabilities.

---

[1]The extra input represents the skip node, which is not needed for `ff` and `ff-hist`, but we add it to make the input consistent across models.

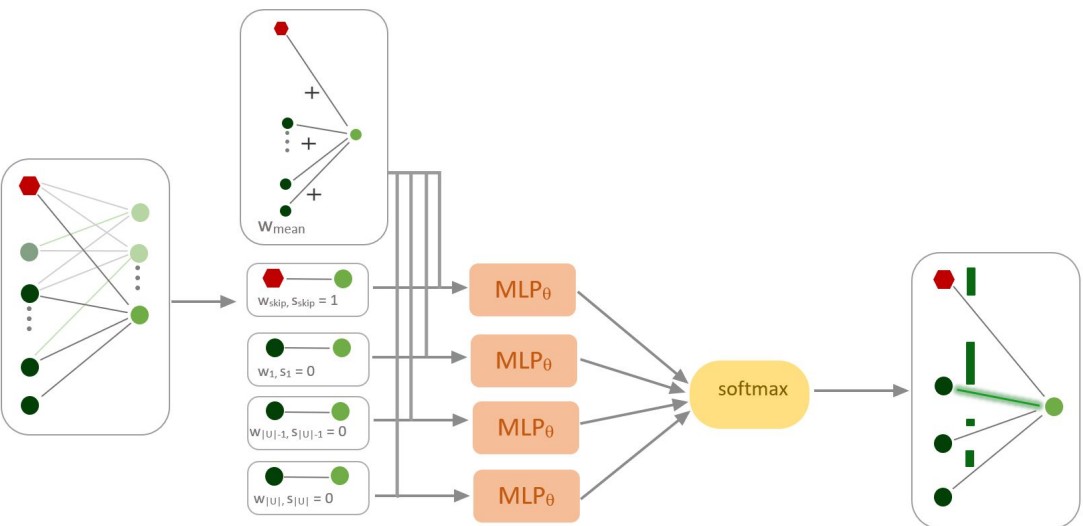

Figure 2: Invariant (`inv-ff`) Architecture. A shared feed-forward neural network takes in node-specific features and outputs a single number for each node in $U$. The outputs are then fed into the softmax function to give a vector of probabilities. The red node represents skipping.

**Invariant Feed-Forward with history** (`inv-ff-hist`): An invariant model, like `inv-ff`, which utilizes historical information. It is important to note that `inv-ff-hist` will only look at historical features of one node at a time, in addition to solution-level features. Therefore, the *node-wise* input is $(w_u, m_u, s_u, w_{mean}, n_t, g_{t,u}, h_t)$.

**Supervised Feed-Forward with history** (`ff-supervised`): To test the advantage of using RL methods, we train `ff-hist` in a supervised learning fashion. In other words, each incoming node is considered a data sample with a target (the optimal $U$ node to match, in hindsight). During training, after all $V$ nodes have arrived, we minimize the cross-entropy loss across all the samples. This setup is equivalent to behavior cloning (Ross & Bagnell, 2010) where expert demonstrations are divided into state-action pairs and treated as i.i.d. samples.

**Graph Neural Network** (`gnn-hist`): In this model, we employ the encoder-decoder architecture used in many combinatorial optimization problems, see (Cappart et al., 2021). At each timestep $t$, the graph encoder consumes the current graph and produces embeddings for all nodes. The decoder feed-forward neural network, which also takes *node-wise* inputs, will take in $(w_u, t/|V|, m_u, s_u, p_t, e_{v_t}, e_u, e_{mean}, e_s)$ where the last four inputs represent the embedding of the incoming node $v_t$, embedding of the fixed node $u$ being considered, mean embedding of all fixed nodes, and mean solution embedding, respectively. Our graph encoder is a MPNN (Gilmer et al., 2017) with the weights as edge features. The mean solution embedding is defined as the sum of a learnable linear transformation of the concatenation of the embeddings of the vertices of the edges in the solution $S$:

$$e_s = \frac{1}{|S|} \sum_{(u,v) \in S} \Theta_e([e_u; e_v]), \tag{1}$$

where ";" represents horizontal concatenation, $\Theta_e$ is a learnable parameter, and $S$ is the set of all matchings made so far. The mean of the embeddings of all fixed nodes is calculated simply as:

$$e_{mean} = \frac{1}{|\bar{U}|} \sum_{u \in \bar{U}} e_u. \tag{2}$$

where $\bar{U} = U \cup \{u_{skip}\}$ and $u_{skip}$ represents the skip node, i.e., matching to this node means skipping. The graph encoder also takes in problem-specific node features if available; see Appendix B.2 for details. The output of the encoder is fed into a feed-forward network which outputs a distribution over available edges.

Table 1: Features used in `ff-hist` and `inv-ff-hist`. $d_u^t$ represents degree of node $u$ at time $t$. $u_{skip}$ represents the skip node, i.e., matching to this node means choosing to skip.

| Feature type | Description | Equation | Size |
|---|---|---|---|
| Graph-Level Features $g_t$ | Average weight per fixed node $u$ up to time $t$ | $\mu_w = \frac{1}{d_u^t} \sum\limits_{\substack{(u,v_i) \in E: \\ v_i \in V, \\ 1 \le i < t}} w_{(u,v_i)}$ | $\|U\| + 1$ |
| | Variance of weights per fixed node $u$ up to time $t$ | $\sigma_w = \frac{1}{d_u^t} \sum\limits_{\substack{(u,v_i) \in E: \\ v_i \in V, \\ 1 \le i < t}} (w_{(u,v_i)} - \mu_w)^2$ | $\|U\| + 1$ |
| | Average degree per fixed node $u$ up to time $t$ | $\frac{1}{t}\|\{(u,v_i) \in E : i \le t\}\|$ | $\|U\| + 1$ |
| Incoming Node Features $n_t$ | Percentage of fixed nodes incident to incoming $v_t$ (For invariant models only) | $\frac{1}{\|U\|}\|\{(u,v_t) \in E : u \in U\}\|$ | $1$ |
| | Normalized step size at time $t$ | $\frac{t}{\|V\|}$ | $1$ |
| Solution-Level Features $h_t$ | Maximum weight in current matching solution | $\max_{e \in S} w_e$ | $1$ |
| | Minimum weight in current matching solution | $\min_{e \in S} w_e$ | $1$ |
| | Mean weight in current matching solution | $\mu_S = \frac{1}{\|S\|}\sum_{e \in S} w_e$ | $1$ |
| | Variance of weights in current matching solution | $\sigma_S = \frac{1}{\|S\|}\sum_{e \in S}(w_e - \mu_S)^2$ | $1$ |
| | Ratio of already matched nodes in $U$ | $\frac{1}{\|U\|}\|\{(u,v) \in S, u \ne u_{skip}\}\|$ | $1$ |
| | Ratio of skips made up to time $t$ | $\frac{1}{t}\|\{(u,v) \in S, u = u_{skip}\}\|$ | $1$ |
| | The normalized size of current matching solution | $p_t = \frac{1}{\|U\|}\sum_{e \in S} w_e$ | $1$ |

The models outlined above are designed based on a set of desirable properties for matching. Table 2 summarizes the properties that are satisfied by each model:

- **Graph Size Invariance**: Training on large graph instances may be infeasible and costly. Thus, it would be ideal to train a model on small graphs if it generalizes well to larger graphs with a similar generating distribution. We utilize normalization in a way to make sure that each statistic (feature) that we compute lies within a particular range, independently of the graph size. Moreover, the invariant architectures allow us to train small networks that only look at node-wise inputs and share parameters across all fixed nodes. It is also worth noting that the invariance property can be key to OBM variants where $U$ is not fixed, e.g., 2-sided arrivals (Dickerson et al., 2018), an application that is left for future work.

- **Permutation Invariance**: In most practical applications, such as assigning jobs to servers or web advertising, the ordering of nodes in the set $U$ should not affect the solution. The invariant architectures ensure that the model outputs the same solution regardless of the permutation of the nodes in $U$. On the other hand, the non-invariant models such as `ff` would predict differently for the same graph instance if the $U$ nodes were permuted.

- **History-Awareness**: A state space defined based on the entire current graph and the current matching will allow the model to learn smarter strategies that reason about the future based on the observed past. Historical and graph-based information within the current graph gives the models an "identity" for each fixed node which may be lost due to node-wise input. Contextual features such as incoming node features $n_t$ (see Table 1) and the ratio of already matched nodes help the learned policies to generalize to different graph sizes and $U$-to-$V$ ratios.

Table 2: Important model characteristics. L: Number of hidden layers, H: Hidden layer size, E: Embedding dimension.

| Model | Graph size Invariance | Permutation Invariance | History Awareness | Node-feature Awareness | Learnable Parameters |
|---|---|---|---|---|---|
| inv-ff | ✓ | ✓ | | | $O(LH^2)$ |
| ff | | | | | $O(LH^2 + |U|H)$ |
| ff-hist | | | ✓ | | $O(LH^2 + |U|H)$ |
| ff-supervised | | | ✓ | | $O(LH^2 + |U|H)$ |
| inv-ff-hist | ✓ | ✓ | ✓ | | $O(LH^2)$ |
| gnn-hist | ✓ | ✓ | ✓ | ✓ | $O(LH^2 + EH + E^2)$ |

- **Node-feature Awareness**: In real-world scenarios, nodes in $U$ and $V$ represent entities with features that can be key to making good matching decisions. For example, incoming nodes can be users with personal information such as age, gender, and occupation. The node features can be leveraged to obtain better matchings. Our GNN model supports node features. Other models can be modified to take such additional features but would need to be customized to the problem at hand.

## 4.3 Training Algorithms

**RL Models**: Because our focus is on flexible modeling of OBM-type problems with deep learning architectures, we have opted to leverage existing training algorithms with little modification. We use the REINFORCE algorithm (Sutton & Barto, 2018), both for its effectiveness and simplicity:

$$\nabla L(\theta|s) = \mathbb{E}_{p_\theta(\pi|s)}[(\mathcal{L}(\pi) - b(s))\nabla \log p_\theta(\pi|s)].$$

To reduce gradient variance and noise, we add a baseline $b(s)$ which is the exponential moving average, $b(s) = M$, where $M$ is the negative episode reward ,$\mathcal{L}(\pi)$, in the first training iteration and the update step is $b(s) = \beta M + (1 - \beta)\mathcal{L}(\pi)$ (Sutton & Barto, 2018).

**Supervised Models**: All incoming nodes are treated as independent samples with targets. Therefore, for a batch of $N$ bipartite graphs with $T$ incoming nodes, we minimize the weighted cross entropy loss:

$$\frac{1}{N \times T}\sum_{i=1}^{N}\sum_{j=1}^{T} loss(p_j^i, t_j^i, c)$$

where $p_j^i$ is output of the policy for graph instance $i$ at timestep $j$, $t_j^i$ is the target which is generated by solving an integer programming formulation on the full graph in hindsight (see Appendix D for details), and $c$ is the weight vector of size $|U| + 1$. All classes are given a weight of 1 except the skipping class which is given a weight of $\frac{|U|}{|V|}$. This is to prevent overfitting when most samples belong to the skipping class, i.e., when $|V| \gg |U|$ and most incoming nodes are left unmatched.

Masking is utilized to prevent all models from picking non-existent edges or already matched nodes.

## 5 Experimental Setup

### 5.1 Dataset Preparation

We train and test our models across two synthetically generated datasets from the Erdos-Renyi (ER) (Erdos & Renyi, 1960) and Barabasi-Albert (BA) (Albert & Barabási, 2002) graph families. In addition, we use two datasets generated from real-world base graphs. The gMission base graph (Chen et al., 2014) comes from crowdsourcing data for assigning workers to tasks. We also use MovieLens (Harper & Konstan, 2015), which is derived from data on users' ratings of movies based on Dickerson et al. (2019). Table 3

summarizes the datasets and their key properties. In our experiments, we generate two versions of each real-world dataset: one where the same fixed nodes are used for all graph instances (gMission, MovieLens), and one where a new set of fixed nodes is generated for each graph instance (gMission-var, MovieLens-var).

To generate a bipartite graph instance of size $|U|$ by $|V|$ from the real-world base graph, we sample $|U|$ nodes uniformly at random without replacement from the nodes on the left side of the base graph and sample $|V|$ nodes with replacement from the right side of the base graph. A 10x30 graph is one with $|U| = 10, |V| = 30$, a naming convention we will adopt throughout. We note that our framework could be used in the non-i.i.d arrival settings. However, the graph generation process depends on the i.i.d assumption, since we sample nodes from the base graph at random.

**Erdos-Renyi (ER):** We generate bipartite graph instances for the E-OBM problem using the Erdos-Renyi (ER) scheme (Erdos & Renyi, 1960). Edge weights are sampled from the uniform distribution $U(0, 1]$. For each graph size, e.g., 10x30, we generate datasets for a number of values of $p$, the probability of an edge being in the graph.

**Barabasi-Albert (BA):** We follow the same process described by (Borodin et al., 2020) for generating preferential attachment bigraphs. To generate a bigraph in this model, start with $|U|$ offline nodes and introduce online nodes $V$ one at a time. The model has a single parameter $p$ which is the average degree of an online node. Upon arrival of a new online node $v \in V$, we sample $n_v \sim Bionomial(|U|, p/|U|)$ to decide the number of the neighbours of $v$. Let $\mu$ be a probability distribution over the nodes in $U$ defined by $\mu(u) = \frac{1+degree(u)}{|U|+\sum_{u \in U} degree(u)}$. We sample offline nodes according to $\mu$ from $U$ until $n_v$ neighbours are selected.

**gMission**: In this setting, we have a set of workers available offline and incoming tasks which must be matched to compatible workers (Chen et al., 2014). Every worker is associated with a location in Euclidian space, a range within which they can service tasks, and a success probability with which they will complete any task. Tasks are represented by a Euclidean location and a payoff value for being completed. We use the same strategy as in (Dickerson et al., 2018) to pre-process the dataset. That is, workers that share similar locations are grouped into the same "type", and likewise for tasks. An edge is drawn between a worker and a task if the task is within the range of the worker. The edge weight is calculated by multiplying the payoff for completing the task with the success probability. In total, we have 532 worker types and 712 task types.

To generate a bipartite graph instance of size $|U|$ by $|V|$, we sample $|U|$ workers uniformly at random without replacement from the 532 types and sample $|V|$ tasks with replacement from $\mathcal{D}$. We set $\mathcal{D}$ to be uniform. That is, the graph generation process involves sampling node from $V$ in the base graph uniformly.

**MovieLens**: The dataset consists of a set of movies each belonging to some genres and a set of users which can arrive and leave the system at any time. Once a user arrives, they must be matched to an available movie or left unmatched if no good movies are available. We have historical information about the average ratings each user has given for each genre. The goal is to recommend movies which are relevant and diverse genre-wise. This objective is measured using the weighted coverage function over the set of genres (see Section 2). Therefore, we must maximize the sum of the weighted coverage functions of all users which have arrived.

The MovieLens dataset contains a total of 3952 movies, 6040 users, and $100,209$ ratings of the movies by the users. As in (Dickerson et al., 2019), we choose 200 users who have given the most ratings and sample 100 movies at random. We then remove any movies that have no neighbors with the 200 users to get a total of 94 movies. These sets of movies and users will be used to generate all bipartite graphs. We calculate the average ratings each user has given for each genre. These average ratings will be used as the weights in the coverage function (see section 2.1). To generate an instance of size $|U|$ by $|V|$, we sample $|U|$ movies uniformly at random without replacement from the 94 movies and $|V|$ users with replacement according to the uniform arrival distribution $\mathcal{D}$. The full graph generation procedure for gMission and MovieLens can be seen in Algorithm 3 of Appendix C.

Table 3: Datasets used for our experiments. $p$ is the average node degree in BA graphs.

| Type | Problem | Base Graph Size | Node Attributes? | Weight Generation |
|---|---|---|---|---|
| ER | E-OBM | | | $w_{(u,v)} \sim U(0,1]$ |
| BA | E-OBM | | | $w_{(u,v)} \sim N(\text{degree}(u), p/5)$ |
| gMission | E-OBM | 532 jobs $\times$ 712 workers | | payoff for computing the task $\times$ the success probability |
| MovieLens | OSBM | 94 movies $\times$ 200 users | ✓ | average ratings each user has given for each genre is used as the weights in the coverage function |

## 5.2 Evaluation

**Evaluation Metric:** We use the *optimality ratio* averaged over the set of test instances. The optimality ratio of a solution $S$ on a graph instance $G$ is defined as $O(S, G) = \frac{c(S)}{OPT(G)}$, where $c(S)$ is the objective value of $S$ and $OPT(G)$ is the optimal value on graph instance $G$, which is computed in hindsight using integer programming; see Appendix D.

**Baselines:** For E-OBM, we compare our models to the `greedy` baseline, which simply picks the maximum-weight edge, and `greedy-rt` (Ting & Xiang, 2014), a randomized algorithm which is near-optimal in the adversarial setting. In an effort to compare to strong tunable baselines, we implemented `greedy-t`, which picks the maximum-weight edge with weight above a dataset-specific threshold $w_T$ that is tuned on the training set. If no weight is above the threshold, then we skip (see Appendix B.4 for details). To best of our knowledge, this is the first application of such a baseline to real-world datasets. For OSBM, we only use `greedy` as (Dickerson et al., 2019) find that it achieves a better competitive ratio than their algorithms when movies cannot be matched more than once and the incoming user can be matched to one movie at a time, which is the setting we study here.

## 5.3 Hyperparameter Tuning and Training Protocol

In a nutshell, around 400 configurations, varying four hyperparameters, are explored using Bayesian optimization (Biewald, 2020) on a small validation set consisting of small graphs (10x30) from the gMission dataset. We have found the models to be fairly robust to hyperparameter values. In fact, most configurations with low learning rates (under 0.01) result in satisfactory performance as seen in Fig 3. The model with the best average optimality ratio on the validation set is selected for final evaluation, the results of which will be shown in the next section. Some hyperparameters are fixed throughout, particularly the depths/widths of the feed-forward networks (2-3 layers, 100-200 neurons), and the use of the ReLU as activation function. Training often takes less than 6 hours on a NVIDIA v100 GPU. Full details are deferred to appendices B.3 and B.1.

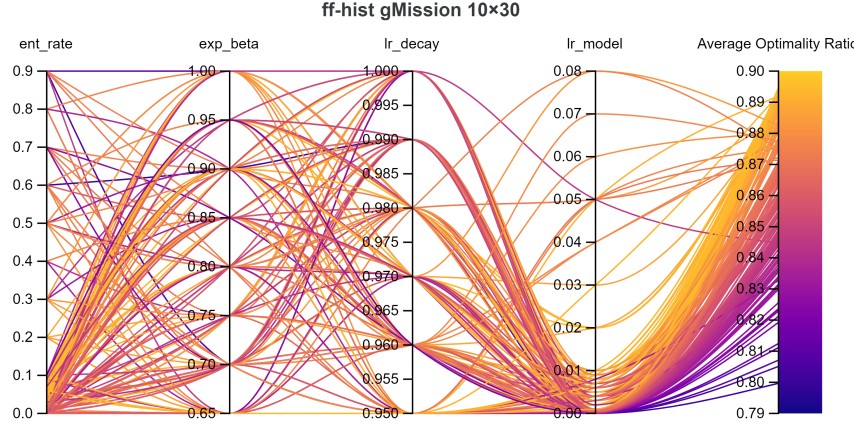

Figure 3: Top 200 hyperparameter tuning results for `ff-hist` on gMission 10x30. Each curve represents a hyperparameter configuration. Lighter color means better average optimality ratio on the validation set.

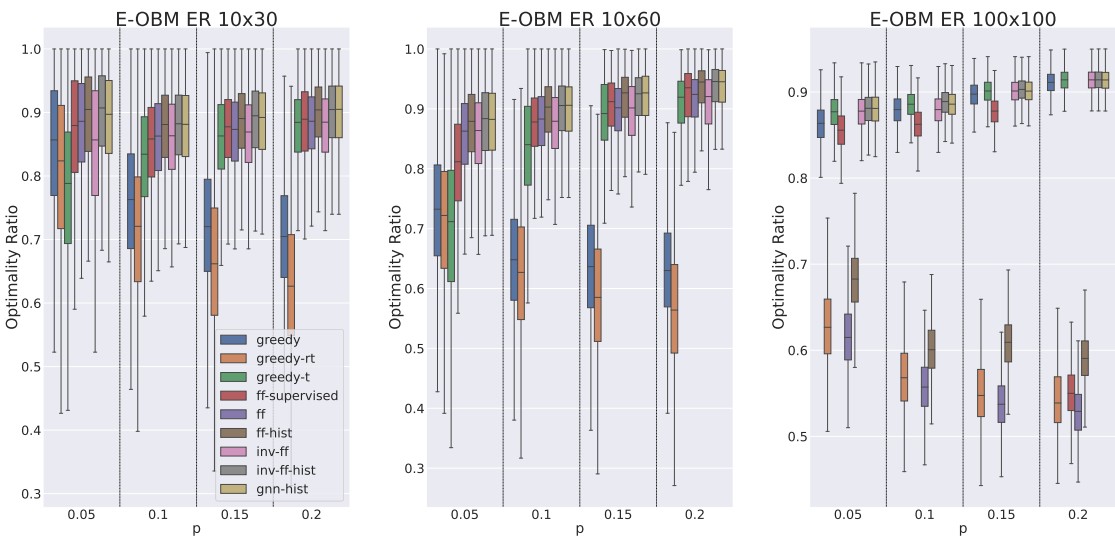

Figure 4: Distributions of the Optimality Ratios for E-OBM on ER graphs. The graph family parameter $p$ is the probability of a random edge existing in the graph.

## 6 Experimental Results

### 6.1 Edge-Weighted Online Bipartite Matching

For E-OBM, we will analyze the performance of the models across ER and BA graphs as well as the gMission datatset. The edges and weights in the ER graphs are generated from a uniform distribution. Thus, ER graphs do not have special graph properties such as the existence of community structures or the occurrence of structural motifs. As a result, the ER dataset is hard to learn from as the models would merely be able to leverage the $|U|$-to-$|V|$ ratio and the density of the graph (the graph family parameter is proportional to the expected node degree in a graph). Unlike ER graphs, explicit structural patterns are found in BA graphs. The BA graph generation process captures heterogeneous and varied degree distributions which are often observed in real world graphs (Barabási & Pósfai, 2016). For example, graphs with many low-degree nodes and a few high-degree nodes occur in practical domains where the rich gets richer phenomenon is witnessed. The BA graph generation process is described in Appendix C. In our experiments, nodes with higher degrees also have higher weights in average. We also study the models under the gMission dataset. Like many real-world datasets, the exact properties of the graphs are unknown. Thus, the models may derive policies based on implicit graph properties. The results will demonstrate that the models have taken advantage of some existing patterns in the dataset.

**Trends in decisions with respect to the $|U|$-to-$|V|$ ratio and graph sparsity:** When $|U| < |V|$, the models outperform the greedy strategies since they learn that skipping would yield a better result in hindsight, despite missing a short-term reward. This is apparent for the 10x30 and 10x60 graphs in Figure 4 for ER and Figure 5 (b) for gMission. To substantiate this and other hypotheses about the behavior of various policies, we use "agreement plots" such as those in Figure 6. An agreement plot shows how frequently the different policies agree with a reference policy, e.g., with a hindsight-optimal solution or with the greedy method. Appendix E includes agreement plots w.r.t. greedy: most disagreements between the learned policies and greedy happen in the beginning but all methods (including greedy) imitate the optimum towards the end, when most actions consist in skipping due to the fixed nodes having been matched already.

Outperforming greedy on 100x100 (3rd plot in Fig. 4) ER graphs is quite a difficult task, as there is not much besides the graph density for the models to take advantage of. Since $|V| = |U|$, skipping is also rarely feasible. Hence, the models perform similarly to `greedy`.

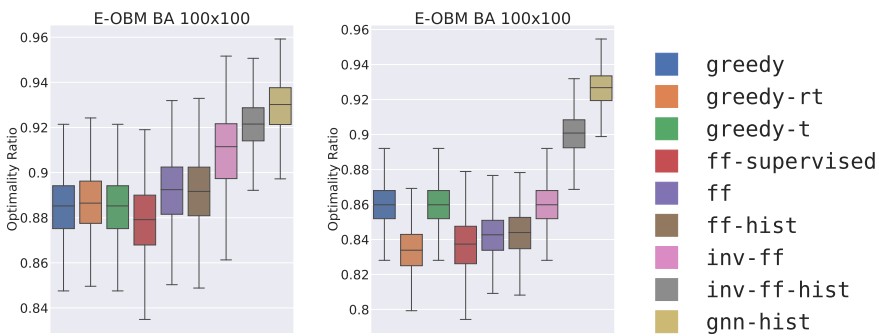

(a) Distributions of the Optimality Ratios for E-OBM on BA graphs with average node degree 5, and weights $w_{(u,v)} \sim N(deg(u), 1)$, and BA with average node degree 15, and weights $w_{(u,v)} \sim N(deg(u), 3)$.

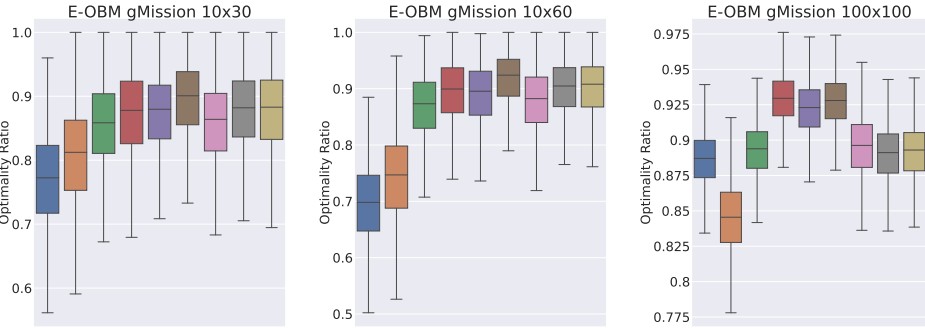

(b) Distributions of the Optimality Ratios for E-OBM on the gMission dataset.

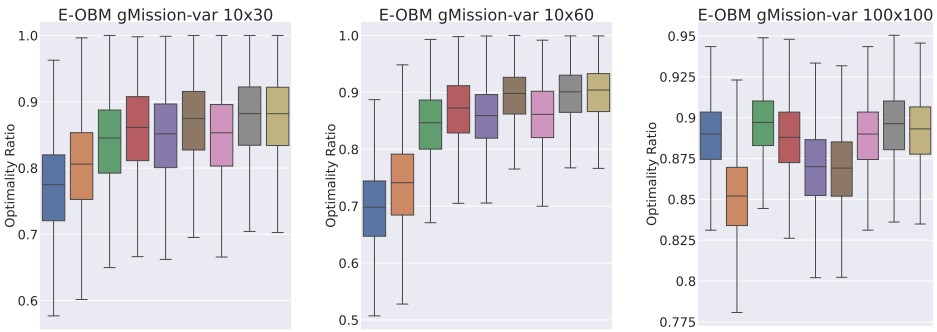

(c) Distributions of the Optimality Ratios for E-OBM on gMission-var.

Figure 5: Distributions of the Optimality Ratios for E-OBM on BA and gMission.

As the ER graphs get denser (moving to the right within the first three plots of Fig. 4), the gap between the models and the greedy baselines increases as there is a higher chance of encountering better future options in denser graphs. Hence, the models learn to skip as they find it more rewarding over the long term. This can be further seen in Fig. 6, where the models agree less with greedy on denser ER graphs. For gMission (right side of Fig. 6), most disagreements happen in the beginning but all models imitate the optimum towards the end when most actions consist in skipping; Appendix E has more agreement plots.

**Model-specific Results:** Models with history, namely `inv-ff-hist` (gray) and `ff-hist` (brown), consistently outperform their history-less counterparts, `ff` and `inv-ff`, across all three datasets (Figure 5).

`inv-ff` receives the same information as `greedy` and performs fairly similar to it on gMission and ER graphs. In fact, `inv-ff` learns to be exactly greedy on gMission 100x100 (see Appendix E). However, `inv-ff` performs better than other non-invariant models on the BA dataset. The ideal policy on BA

Agreement with Optimal for ER 10x30    Agreement with Optimal for gMission 10x30

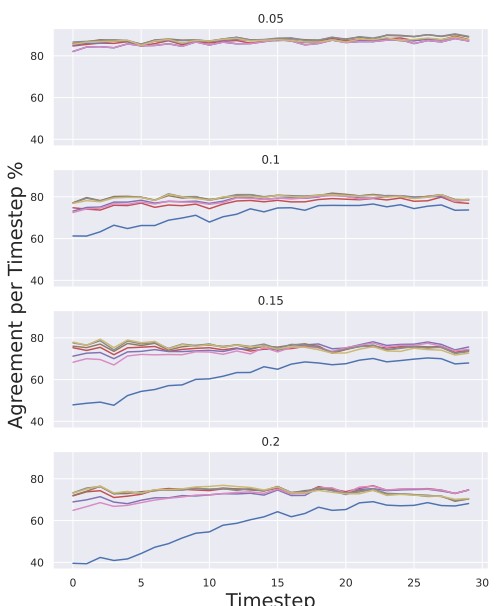
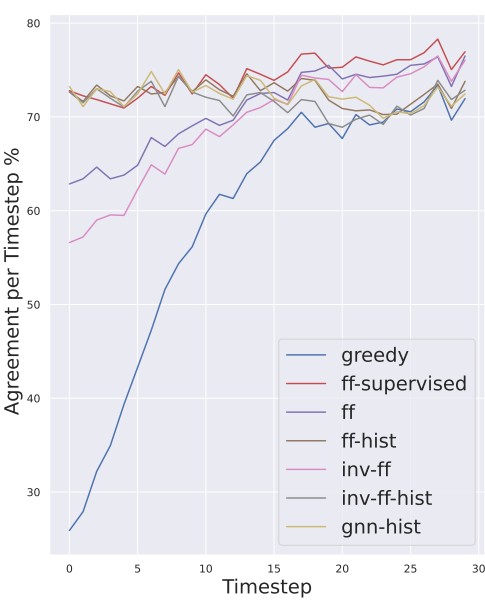

Figure 6: Percent agreement with the optimal solution per timestep. A point (timestep $t$, agreement $a$) on this plot can be read as: at timestep $t$, this method makes the same matching decision as the optimal solution on $a\%$ of the test instances.

graphs would discover that matching to a high-degree node is not wise since the node will likely receive a lot more edges in the future. Similarly, matching to a low-degree node is desired since the node will probably not have many edges in the future. The node-wise reasoning of the invariant models is effective at learning to utilize such node-specific graph property and following a more feasible policy. Armed with historical context, `inv-ff-hist` outperforms all other models on BA graphs (Fig. 5a).

The best performance on ER and gMission is achieved by `ff-hist` since the model gets all information (weights) at once (Fig. 5). However, when $U$ nodes are permuted, `inv-ff-hist` vastly outperforms `ff-hist`, as shown in Appendix G.

`ff-supervised` performs well but not as good as RL models since supervised learning comes with its own disadvantages, i.e, overfitting when there are more skip actions than match, and being unable to reason sequentially when it makes a wrong decision early on. The latter is a well-known fatal flaw in behavior cloning, as observed by Ross & Bagnell (2010) and others.

`greedy-t` performs well compared to `greedy`, which shows the advantage of strategically skipping if weights do not pass a tuned threshold. However, it is still outperformed by the learned models, especially on BA graphs where the graphs exhibit explicit patterns and gMission 100x100.

In general, the choice of the best model is dependent on the problem, but we provide some empirical evidence on how this choice should be made. The invariant models with history (`inv-ff-hist`, `gnn-hist`) are the best performing models and most recommended to be used in practice as they are invariant to $|U|$, can support more general OBM variants such as 2-sided arriving nodes, and can take advantage of node/edge features. For settings where $|U|$ is always fixed (e.g., scheduling jobs to servers), `ff-hist` is the best as it can see all arriving edges at once and takes advantage of history.

**Some advantages to using invariant models:** We also experiment with a variation on the gMission dataset, where a new set of fixed nodes is generated for each graph instance. We see the same pattern as in gMission (where the same fixed nodes in $|U|$ existed across all instances), except non-invariant models

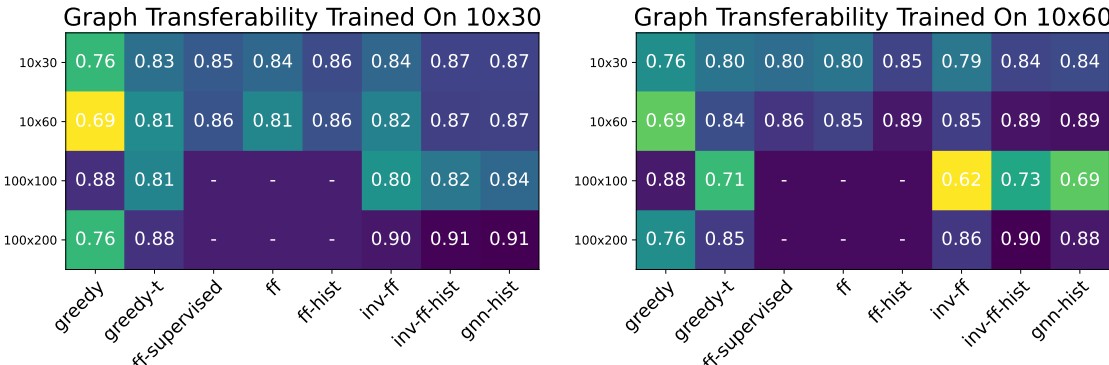

Figure 7: Graph Transferability on gMission-var: Average optimality ratios for models trained on graphs of size 10x30 (left) & 10x60 (right) and tested on graphs of different sizes. Missing values are denoted with a dash for models that are not invariant to the number of $U$ nodes of the training graphs.

degrade substantially for 100x100. This is because the input size increased substantially but the model's hidden layer sizes were kept constant. The same issue is seen for BA graphs. A significant disadvantage of non-invariant models is that they are not independent of $|U|$, so model size needs to be increased with $|U|$. Invariant models are unaffected by this issue as they reason *node-wise*. We notice that this problem is not seen in gMission. One explanation for this is that fixed nodes are the same across all instances so models have a good idea of the weight distribution for each fixed node which is the same during testing and training. Therefore, even though the model size is not increased as the input size increases, the models can in some sense "guess" the incoming weights and so do not need as much of an increase in capacity. Once again, models with history display better performance as history helps the models build a better "identity" of each fixed node as more nodes come in even if never seen during training.

**Do models trained on small graphs transfer to larger graphs?** In Fig. 7, we train all models on 10x30 and 10x60 graphs separately and test their transferability to graphs with different $|U|$-to-$|V|$ ratios up to size 100x200. `gnn-hist` and `inv-ff-hist` perform especially well on graphs with similar $|U|$-to-$|V|$ ratio. For 100x100 graphs, `inv-ff` and `greedy-t` perform poorly as they do not receive any features that give them context within a new graph size such as number of available fixed nodes.

### 6.2   Online Submodular Bipartite Matching (OSBM)

The inherent complexity of the OSBM problem and the real-world datasets provide a learning-based approach with more information to leverage. As such, the models tend to discover policies which do significantly better than `greedy` as shown in Fig. 8.

The benefit of RL models is apparent here when compared to `ff-supervised`, particularly for 10x30 and 94x100 graphs. The relative complexity of OSBM compared to E-OBM will require the model to be more generalizable as the reasoning involved is more complex and mere imitation is not enough. `ff-supervised` also underperforms because the edge weights depend on the current solution and can change on the same graph instance if previous matches are different, causing a great mismatch with the training data.

A similar trend to E-OBM results is observed here: models with history outperform their history-less counterparts. The context provided by history is particularly helpful as the edge weights depend on previous matches. Furthermore, we notice that `gnn-hist` has the best performance on 10x30 and 94x100 graphs as `gnn-hist` is the only model that uses user attributes as node features.

We witness the same issue seen in gMission-Var (5). The non-invariant models degrade on 94x100 graphs due to having the same number of hidden layer despite processing larger graphs. The invariant models

remain unaffected by the graph size. Interestingly, the invariant models even slightly outperform their non-invariant counterparts on 10x30 and 10x60 MovieLens-var.

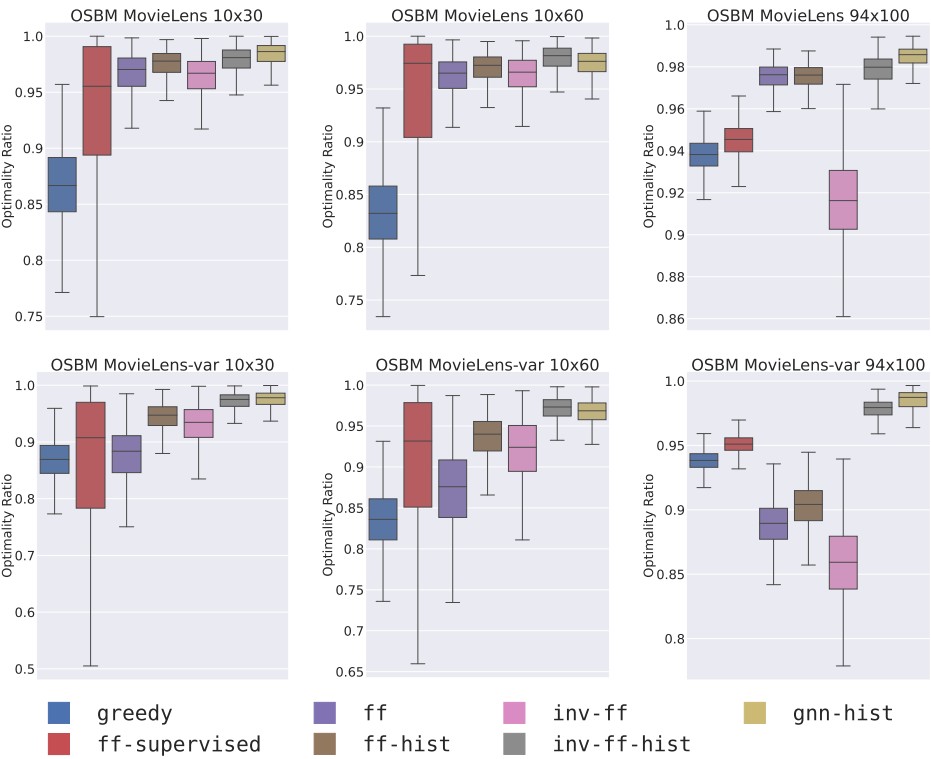

Figure 8: Distributions of the Optimality Ratios for OSBM on three graph sizes for MovieLens and MovieLens-var. Higher is better.

## 7 Conclusion and Future Work

Through extensive experiments, we have demonstrated that deep reinforcement learning with appropriately designed neural network architectures and feature engineering can produce high-performance online matching policies across two problems spanning a wide range of complexity. In particular, we make the following concluding observations:

- A basic reinforcement learning formulation and training scheme are sufficient to produce good learned policies for online matching, are typically not sensitive to the choice of hyperparameters, and are advantageous compared to a supervised learning approach;

- Compared to greedy policies, RL-based policies are more effective, a result that can be partially explained by a stronger agreement with the optimal solution (in hindsight) in the early timesteps of the process when greedy is too eager to match. RL policies tend to perform particularly well when trained and tested on dense graphs or ones with a strong structural pattern;

- Models that are invariant to the number of nodes are more advantageous than fully-connected models in terms of how well they generalize to test instances that are slightly perturbed compared to the training instances, either in graph size or in the identities of the fixed nodes;

- Feature engineering at the node and graph levels can help model the history of the matching process up to that timestep, resulting in improved solutions compared to models that use only weight information from the current timestep;

- Graph Neural Network models are a viable alternative to feed-forward models as they can leverage node features and their dependencies across nodes more naturally.

Future avenues of research include:

- A more extensive experimental analysis of different RL training algorithms beyond basic policy gradient;

- Extensions to new real-world datasets with rich node and edge features that could benefit even more from highly expressive models such as GNNs;

- Extensions to other online combinatorial optimization problems, which can leverage our framework, models, and code as a starting point.

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

## A   Implementation Details

All environments are implemented in Pytorch (Paszke et al., 2019). We use NetworkX (Hagberg et al., 2008) to generate synthetic graphs and find optimal solutions for E-OBM problems. Optimal solutions for OSBM problems are found using Gurobi (Gurobi Optimization, LLC, 2021); see Appendix D. Pytorch Geometric (Fey & Lenssen, 2019) is used for handling graphs during training and implementing graph encoders. Our code is attached to the submission as supplementary material.

## B   Training and Evaluation

### B.1   Training Protocol

We train our models for 300 epochs on datasets of 20000 instances using the Adam optimizer (Kingma & Ba, 2015). We use a batch size of 200 except for MovieLens, where batch size 100 is used on graphs bigger than 10x60 due to memory constraints.

Training often takes less than 6 hours on a NVIDIA v100 GPU. `gnn-hist` takes less than a day to train on small graphs but consumes more time for bigger graphs and more complicated environments such as MovieLens 94x100. This is due to re-embedding the graph at every timestep which consumes more memory and computation as the graph size grows. We believe this can be improved with more efficient embedding schemes for dynamic graphs (Kazemi et al., 2020) but leave this for future work.

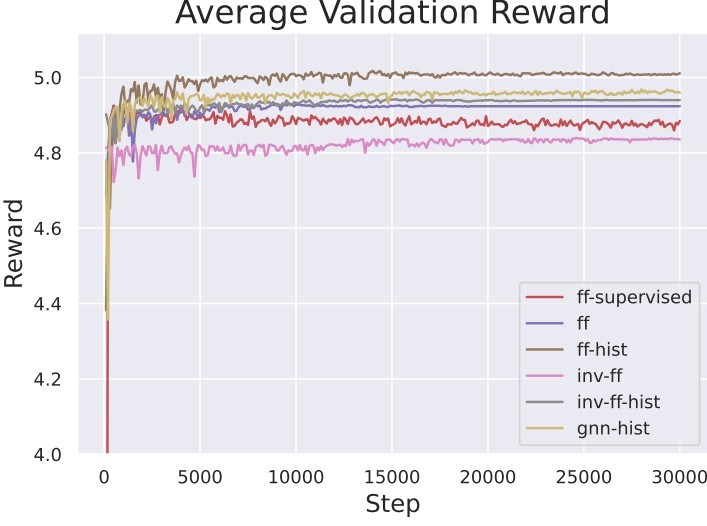

Figure 9: Average Validation reward during training on gMission 10x30. Note that `ff-supervised` starts with a reward of approximately 1.9 at batch 0.

### B.2   Node Features

Since `gnn-hist` supports node features by default, we leverage this property in order for incoming nodes to have "identities" that can be helpful for the learning of effective strategies.

For the E-OBM problem, the ER and gMission datasets do not have any helpful node attributes that are not encoded in the edge weights. Therefore, we only provide node attributes that help the encoder differentiate between different types of nodes. That is, the skip node will get node feature $-1$, any fixed node $i$ will get feature $(j_i)$ which is 1 if $i$ is already matched and 0 otherwise, and incoming nodes have feature 2. These features simply help the model differentiate between incoming nodes, fixed nodes, and the node that represents the skipping action.

In the original MovieLens dataset, the incoming nodes are users while the fixed nodes are movies, both of which have helpful features. A fixed node $i$ has feature vector $(j_i, g_i)$ where $g_i$ is a binary vector that represents the genres which the movie belongs to. The skip nodes will have feature vector $\vec{0}$. Incoming users have attributes (gender, occupation, age), each of which is mapped to a number and normalized to between 0 and 1. See Table 4 for details.

Table 4: User features in the MovieLens dataset.

| Attribute | Categories | Feature Range |
|---|---|---|
| Gender | Male, Female | $0 \le i \le 1$ |
| Age | <ul><li>Under 18</li><li>18-24</li><li>25-34</li><li>35-44</li><li>45-49</li><li>50-55</li><li>56+</li></ul> | $0 \le i \le 6$ |
| Occupation | <ul><li>Other</li><li>academic/educator</li><li>artist</li><li>clerical/admin</li><li>college/grad student</li><li>customer service</li><li>doctor/health care</li><li>executive/managerial</li><li>farmer</li><li>homemaker</li><li>K-12 student</li><li>lawyer</li><li>programmer</li><li>retired</li><li>sales/marketing</li><li>scientist</li><li>self-employed</li><li>technician/engineer</li><li>tradesman/craftsman</li><li>unemployed</li><li>writer</li></ul> | $0 \le i \le 20$ |

### B.3 Hyperparameter Tuning

We tune 4 training hyperparameters for each RL model using a held-out validation set of size 1000. The hyperparameters are the learning rate, learning rate decay, exponential decay parameter, and entropy regularization rate. For the `supervised-ff` model, only the learning rate and learning rate decay are tuned. Hyperparameters are optimized using the Bayesian search method (Snoek et al., 2012) in the Weights & Biases library (Biewald, 2020) with default parameters. We conduct around 400 runs for each model.

All models are tuned on small bipartite graphs ($10 \times 30$) from the gMission dataset; the same hyper-parameters are used for bigger graphs at evaluation time. We also use the same hyper-parameters for all datasets. We have found the models to be fairly robust to different hyperparameters. As can be seen in Figure 3, most configurations with low learning rates (under 0.01) result in satisfactory performance.

**Fixed Hyperparameters**: The `ff` and `ff-hist` models have 3 hidden layers with 100 neurons each. `inv-ff` and `inv-ff-hist` have 2 hidden layers of size 100. The `gnn-hist`'s decoder feed-forward neural

Table 5: Hyperparameter Grid

| Hyperparameter | Range |
|---|---|
| Learning Rate | $\{j \times 10^{-i} \mid 2 \leq i \leq 6, 1 \leq j \leq 9\}$ |
| Learning Rate Decay | $\{1.0, 0.99, 0.98, 0.97, 0.96, 0.95\}$ |
| Exponential Decay $\beta$ | $\{1.0, 0.95, 0.9, 0.85, 0.8, 0.7, 0.65, 0.6\}$ |
| Entropy Rate $\gamma$ | $\{j \times 10^{-i} \mid 1 \leq i \leq 4, 1 \leq j \leq 9\}$ |

---

**Algorithm 1** `greedy-rt`

---

1: Choose an integer $K$ uniformly at random in the set $N = \{0, 1, \ldots, \lceil \ln(w_{max} + 1) \rceil - 1\}$
2: Set $\tau = e^K$
3: **while** *a new vertex $v \in V$ arrives* **do**
4:      $A = \{u \mid u$ *is $v$'s unmatched neighbor in $U$ and* $w_{(u,v)} \geq \tau\}$
5:      **if** $A = \phi$ **then**
6:          *leave $v$ unmatched*
7:      **else**
8:          *match $v$ to an arbitrary vertex in $A$*

---

network has 2 hidden layers of size 200 and the encoder uses embedding dimension 30 with one embedding layer. All models use the ReLU non-linearity.

Each RL model is tuned on 4 hyperparameters, as seen in Table 5, using a held-out validation set of size 1000. The figure below shows the top 200 hyperparameter search results for `ff-hist`. Each curve represents a hyperparameter configuration. Evidently, most configurations with small learning rates result in a high average optimality ratio. Other models also show similar results in being insensitive to minor hyperparameter changes. All experiments are done with a constant random seed. Unlike other RL domains, we have not found the models to be sensitive to the seed and never had to restart training due to a bad run. This could be explained by the fact that the transitions in OBM are deterministic regardless of the action (skip or match). Therefore, the MDP is not highly stochastic.

### B.4 Evaluation

`greedy-rt` **baseline**: As shown in Algorithm 1, `greedy-rt` works by randomly picking a threshold between $e$ and $e^{\lceil \ln(w_{max}+1) \rceil}$, where $w_{max}$ is the maximum possible weight. When a new node comes in, we arbitrarily pick any edge whose weight is at least the threshold, or skip if none exist. Surprisingly, this simple strategy is near-optimal in the adversarial setting, with a competitive ratio of $\frac{1}{2e\lceil \ln(w_{max}+1) \rceil}$ (Ting & Xiang, 2014).

Since `greedy-rt` does not support weights between 0 and 1, we re-normalize the edge weights in all E-OBM datasets. For ER and BA graphs, we divide all weights by the minimum weight in the dataset. For gMission graphs, we re-normalize by multiplying all weights by the maximum weight in the original dataset.

`greedy-t` **baseline**: Gaining intuition from `greedy-rt`, we implement a baseline where the threshold is tuned rather than randomly picked. That is, we find a threshold $w_T \in \{0.01, 0.02, \ldots, 1.\}$ that achieves the best average reward on the training set. Then, we use $w_T$ as fixed threshold for all test graphs. See Algorithm 2 for pseudo-code.

## C  Dataset Generation Details

We provide high-level pseudocode for graph generation in Algorithm 3. $K$ is the number of fixed nodes, $T$ is the time horizon and also the number of incoming $V$ nodes, $N$ is the number of instances to be generated, and "type" can be set to "var" to ask for graphs with varying sets of fixed nodes, with the default being to use the same set of fixed nodes.

---

**Algorithm 2** `greedy-t`

---

1: **Input** $w_T$.
2: **while** *a new vertex $v \in V$ arrives* **do**
3:     $A = \{u | u$ *is $v$'s unmatched neighbor in $U$ and $w_{(u,v)} \geq w_T\}$*
4:     **if** $A = \phi$ **then**
5:         *leave $v$ unmatched*
6:     **else**
7:         *Match $v$ with the maximum-weighted edge to a node in $A$*

---

**Algorithm 3** Graph Generation

---

1: **procedure** GENERATE($K$, $T$, $\mathcal{D}$, $G(U^*, V^*, E^*)$, type, $N$)
2:     $D = \{\}$
3:     **if** type != "var" **then**             ▷ if all graphs should have same fixed nodes
4:         $U = u_1, \ldots, u_K \sim Uniform(U^*)$    ▷ Sample K fixed nodes without replacement from base graph
5:     **while** $i < N$ **do**                ▷ Generate $N$ graphs
6:         **if** type = "var" **then**
7:             $U = u_1, \ldots, u_K \sim Uniform(U^*)$         ▷ Re-sample for every graph
8:         $V, E = \{\}, \{\}$
9:         **while** $j < T$ **do**             ▷ Add $T$ nodes to $V$
10:             $v \sim \mathcal{D}(V^*)$       ▷ Sample according to arrival distribution $\mathcal{D}$ from base graph
11:             $e = \{(u,v) : u \in U\}$
12:             **if** $e = \phi$ **then**
13:                 Go to Step 10         ▷ Re-sample if incoming node has no neighbors
14:             $V = V \cup \{v\}$
15:             $E = E \cup e$
16:             $j \mathrel{+}= 1$
17:         $D = D \cup \{G(U, V, E)\}$           ▷ Add graph instance to dataset
18:         $i \mathrel{+}= 1$
      **return** $D$             ▷ Return $N$ graphs of size $K$ by $T$

---

# D   Finding Optimal Solutions Offline

To find the optimal solutions for E-OBM, we use the `linear_sum_assignment` function in the SciPy Library (Jones et al., 2001–). For OSBM (with the coverage function), we borrow the IP formulation from (Dickerson et al., 2019) defined below, where $[g]$ is the set of genres and $z$ is used to index into each genre. Every edge is associated with a binary feature vector $q_{(u,v)}$ of dimension $g$:

$$\text{Maximize} \sum_{v \in V} \sum_{z \in [g]} w_{zv} \gamma_{zv}$$

$$\text{Subject to} \sum_{u \in \text{Ngbr}(v)} x_{uv} \leq r_v \quad , \forall v \in V$$

$$\sum_{v \in \text{Ngbr}(u)} x_{uv} \leq 1 \quad , \forall u \in U$$

$$\gamma_{zv} \leq \sum_{(u,v) \in E : q_{(u,v)}[z]=1} x_{(u,v)} \quad , \forall z \in [g], v \in V$$

$$\gamma_{zv} \leq 1 \quad , \forall z \in [g], v \in V$$

$$x_{uv} \in \{0, 1\}$$

## E  Agreement Plots

In Figure 10a, `greedy` is closer to optimal for sparse graphs but gets increasingly different as the graphs get denser. This suggests that sparse graphs require a more greedy strategy as there not many options available in the future, while dense graphs with more incoming nodes than fixed nodes require a smarter strategy as the future may hold many better options.

For gMission 100x100, invariant models are closer to `greedy` in the beginning only. Non-invariant models learn very different strategies from `greedy`. Interestingly, while `ff-supervised` does not have the best performance on gMission 100x100, it is the closest to optimal in Figure 10c. This is because the supervised model is learning to copy the optimal actions but lacks the sequential reasoning that RL models possess. It is often the case that there are many different solutions besides optimal that give high reward, so RL models may not necessarily learn the same strategy as `ff-supervised`.

## F  Adwords

In order to demonstrate that our RL framework can also tackle other variations of OBM, we test the performance of our models on the Adwords Mehta et al. (2005) problem. Adwords is a variation of OBM with applications in real-time ad allocation. Each vertex $u \in U$ has a budget $B_u$, and each edge $(u, v)$ has a bid (weight) $bid_{uv}$. Upon matching the edge $(v, u)$, the node $u$ depletes $bid_{uv}$ amount of its budget. When a vertex is out of budget ($B_u = 0$), then the vertex becomes unavailable. The goal is to maximize the total amount of budget spent.

To give the hist-models (`inv-ff-hist`, `ff-hist`) more context about the problem state, we input an additional vector of the remaining and original budgets of nodes $(r_0, r_2, \ldots r_{|U|}, B_0, B_2, \ldots, B_{|U|})$, where $r_i$ is the remaining budget of node $i$ at time t [2].

We train and test our models on two hard distributions, Thick-z and Triangular, used in the work by Zuzic et al. (2020). The instances in the datasets are generated by permuting the nodes in $U$, all edges have the same $bid$ sampled from $U[0.1, 0.4]$. The budget of each fixed node is $bid\frac{|V|}{|U|}$. The order in which the nodes in $V$ arrive is the same across all instances.

We find that `ff` and `inv-ff` converge to one of the existing policies (greedy and MSVV), or outperform them by a small margin. This result is coherent with the findings in Zuzic et al. (2020). However, we can see that `inv-ff-hist` and `ff-hist` outperform the baselines on thick-z graphs. In particular, `inv-ff-hist` is able to achieve 0.91 average optimality ratio.

It is noteworthy that, in such hard graph datasets, only the permutation of the U nodes and $bid \sim U[0.1, 0.4]$ change for each instance. If one discovers the real identities (order) of the U nodes, then achieving the optimal matching is trivial Zuzic et al. (2020). Therefore, the historical data coupled with the invariance property gives `inv-ff-hist` better inductive bias to discover the identities and achieve near-optimal matching performance.

In traditional RL applications, a sub-optimal action can result in the agent entirely changing its trajectory ahead. In the E-OBM setting, however, a sub-optimal action is not guaranteed to affect all the future actions (edge selections), especially on sparse graphs. In Adwords, the impact of a single sub-optimal action is even less significant on the total budget spent (and the cumulative reward). That is, a sub-optimal action is very unlikely to affect all the future actions (edge selections) in the future. Hence, there will be less need to be strategic as the penalty for a sub-optimal action is negligible. Thus, discovering a new policy on real-world graphs will be quite hard. In general, the existing greedy algorithms for Adwords perform quite well and are hard to beat (especially when the bids are significantly smaller than budgets).

---

[2]Models without history are only given the current budget.

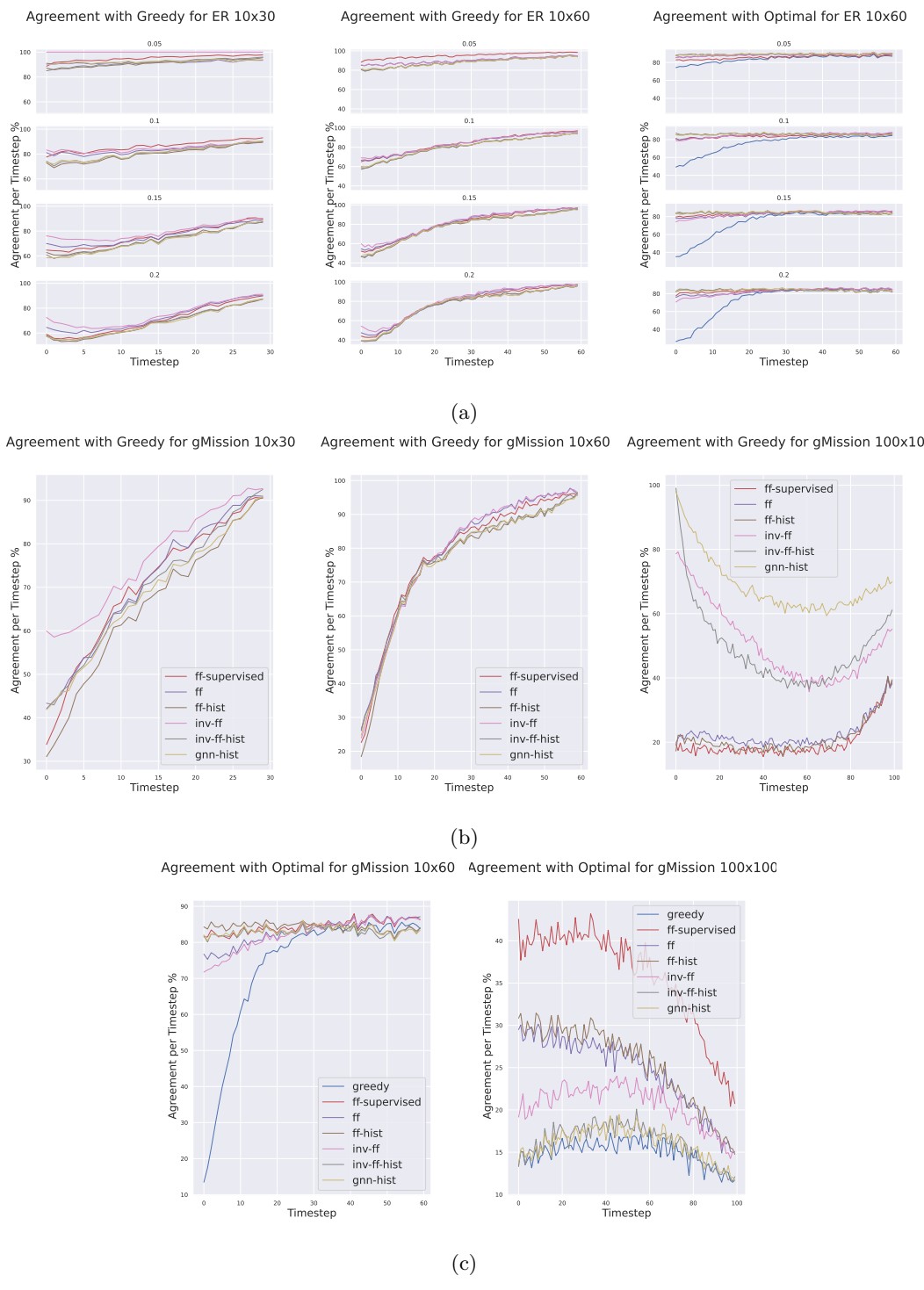

(a)

(b)

(c)

Figure 10: Percent agreement with the optimal solution per timestep. A point (timestep $t$, agreement $a$) on this plot can be read as: at timestep $t$, this method makes the same matching decision as the optimal solution on $a\%$ of the test instances.

Table 6: Mean/std optimality ratio on 2 hard Adwords datasets Zuzic et al. (2020). All graphs in the training and test set have the same structure except the fixed nodes are permuted and the bids are sampled uniformly between 0.1 and 0.4 for all edges.

| | Thick-z | | Triangular | |
|---|---|---|---|---|
| | 10x60 | 10x100 | 10x60 | 10x100 |
| greedy | $0.59 \pm .03$ | $0.58 \pm .02$ | $0.66 \pm .03$ | $0.66 \pm .02$ |
| MSVV | $0.7 \pm .01$ | $0.7 \pm 0$ | $0.66 \pm 0$ | $0.66 \pm 0$ |
| ff | $0.7 \pm .08$ | $0.7 \pm .09$ | $0.65 \pm .06$ | $0.65 \pm .06$ |
| ff-hist | $0.77 \pm .07$ | $0.71 \pm .09$ | $0.65 \pm .06$ | $0.65 \pm .06$ |
| inv-ff | $0.7 \pm .01$ | $0.7 \pm 0$ | $0.66 \pm 0$ | $0.66 \pm 0$ |
| inv-ff-hist | $0.91 \pm 0$ | $0.91 \pm 0$ | $0.66 \pm 0$ | $0.66 \pm 0$ |

## G  Permutation Invariance

In Figure 11, we show the results on gMission-perm where we train all models on the gMission dataset and test on the same dataset but with the fixed nodes permuted for each graph instance independently. We can see that the performance of the non-invariant models degrades significantly compared to the gMission results. The invariant models are unaffected by the permutation as they receive *node-wise* input. `ff-hist` is less affected by this permutation in the 10x60 plot, this suggests that historical features help the model learn "identities" for each fixed node even if the nodes are permuted at test time. However, more incoming nodes need to be observed for the features to be statistically significant.

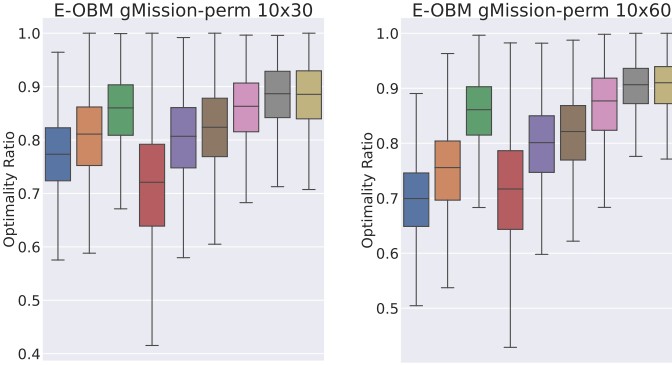

Figure 11: Distribution of Optimality Ratios for gMission-perm: gMission with Fixed Nodes Permuted at Test Time

