# OpenReview forum: "Deep Policies for Online Bipartite Matching: A Reinforcement Learning Approach"
_TMLR — Accepted by TMLR_

### Review · Reviewer_Umxe · 2022-08-01

**Summary Of Contributions:**

Online bipartite matching is a well-known problem in combinatorial optimization. The idea is that nodes from one side of a bipartite graph arrive online one at a time, each with some outgoing edges of different weights. One must decide upon seeing each node whether to irrevocably match it to another node, or to skip it. The goal is to maximize the total edge weight of the final matching.

This problem is heavily studied from a theoretical perspective, often looking for worst-case guarantees. However, as the authors observe, it is extremely natural to model this problem as an MDP. Then one might ask whether, given access to some simulation or other source of information about node arrivals, reinforcement learning can learn a good policy. This is exactly what the authors do.

Modeling the problem from the perspective of OBM, the authors choose to focus on the "unknown IID setting", which is previously studied in the literature and also conveniently seems to be amenable to coding up RL simulation environments from existing bipartite matching datasets.

Modeling the problem from the perspective of RL, the authors have a fairly straightforward MDP, but they point out an important property, which is that "uncertainty is exogenous". In other words, the result of the agent's action (choose to match an edge) is completely predictable and has predictable immediate reward -- stochasticity comes in only from the random node arrivals, which are independent of the agent's actions.

Given this model, they present a thorough examination of how to use RL to solve this problem. They test on a few standard random graph models and use a simple REINFORCE w/ baseline policy optimization approach. They present a thorough empirical study of many different neural network architectures, in particular covering important properties like permutation invariance and ability to generalize to unseen graph sizes.


**Broader Impact Concerns:**

There are no broader impact concerns.

**Requested Changes:**

- Weak suggestions:
  - Everyone uses transformers for everything nowadays, and they're also permutation equivariant and can generalize to new sizes. This might be an interesting thing to try.
  - All other suggestions I thought of were in fact already mentioned in authors' last list of bullet points.

A critical request: please discuss initialization/random seeds explicitly, for both network parameters and environment randomness. This is always a big problem in deep RL. Based on the way the results are structured here, the robustness to hyperparameters, and the repeated trials, I don't think that "tuning the random seed" is a likely explanation for good performance. But I do believe seeds are essential to mention briefly in the paper.

**Strengths And Weaknesses:**

The main empirical section is very carefully done. In terms of experimental setup, they choose reasonable baselines to investigate -- some theoretical baselines, as well as behavioral cloning (which they find does not perform as well as true RL). Testing against optimal hindsight decision-making is also good, and comparing not just performance but decisions chosen against hindsight gives some useful information. More broadly, it's good that the authors actually look to see what their learned policies are doing. The hyperparameter tuning also seems to be far more thorough than is usual.

In terms of the choice of network architectures, the authors seem to explore all the design choices that matter. Intuitively, this is a problem where invariance should be valuable -- the authors find that it is. It also makes sense that one would hope for generalization to multiple graph sizes, which the authors find is possible. I can believe that giving access to history features matters in practice, but it seems sort of odd as this is an MDP.

The major weakness is that this is a completely reasonable choice of problem, but it is also not that profound. It is not surprising that OBM can be formulated as an MDP, and that given such an MDP, standard deep RL techniques can find good solutions. On the other hand, it's good that someone actually bothered to do this, and did a careful job of it.

---

> ### Author Response · Authors · 2022-08-28
> **Reply to Reviewer Umxe**
>
> Thank you very much for your thorough review!
>
> We believe using transformers would be an interesting future direction. However, unlike domains where transformers have shown strong results (such as language modeling), the complexity of online combinatorial problems lies within the uncertainty about future rather than complexity in data (as is the case with many other supervised learning tasks, such as NLP). We suspect that transformers might be a little excessive for our purpose but will consider trying this out in future work.
>
> Thank you for raising the point about random seeds! In our experiments, we use a random seed for all our experiments. Unlike other RL domains, we have not found the models to be sensitive to the seed and never had to restart training due to a bad run. This could be explained by the fact that the transitions in OBM are deterministic regardless of the action (skip or match). Details about the random seed have been added to Appendix B.3.

---

### Review · Reviewer_6WVx · 2022-08-02

**Summary Of Contributions:**

This paper studies the online bipartite matching problem, where one side of the nodes are pre-selected and fixed; the other side of the nodes are arriving in an online fashion. The algorithm needs to make a matching or choose to skip every time a new node arrives. The goal is the maximize the total number of matches (or weighted matches) within a given time horizon T.

The paper considers the setting where the nodes arriving are iid samples drawn from an unknown fixed distribution, and proposes a reinforcement learning framework for this problem. Based on this MDP framework the authors evaluated a set of neural network architectures, feature representations, and empirically evaluate the models on two online matching problems: Edge-Weighted Online Bipartite Matching and Online Submodular Bipartite Matching.

In the evaluation part, the authors compared to a few greedy-based baselines.

**Broader Impact Concerns:**

I did not find major ethical concerns about this work.

**Requested Changes:**

Critical:
- please improve the writing and presentation on motivation and related works
- please add comparisons to stronger baselines in prior works
- please enhance the experimental results with larger scale data / rnn-type models

Minor:
- fix typos
- please add discussion about the dependence / robustness of the RL models towards the iid assumption
- please improve the presentation and arrangements of the figure (e.g. figure 1, see comments above)


**Strengths And Weaknesses:**

# Strengths

- the problem setting is very clear
- the MDP framework and the application of RL algorithm to me seem to be very natural and reasonable solution to the problem
- the experimental section was rather detailed, the break down of the features used; the results of the hyper parameter tuning etc were nicely presented

# Weaknesses

- the related work section is not comprehensive enough and some references and details are missing. For example, the author noted that "Many traditional algorithms under the KIID setting aim to overcome this challenge by explicitly approximating the distribution" but there were no references for that. I suggest these related references to be added.

- the writing could be improved much to make the paper easier to follow, especially the intro / related work sections, see below for more details

- I feel that the baselines that the authors compared with were fairly simple, e.g the greedy algorithm. It seems to be more fair to compare the RL models to other predictive-type models, e.g the Q-learning approach and other RL approaches mentioned in the related work section. I think stronger baselines comparisons are needed

- It was not very clear to me where the RL framework and solution is depending on the "iid" assumption. Why can't such an algorithm work with non iid, or adversarial node samples? Can one stress test the robustness of the models toward non-iid samples?

- in the models that were used, there were a few (e.g. ff with history) that tried to include historical information into the model. How does a RNN base model perform compared to the models that were used in the paper?

- the scale of the experiments was a bit limited.  e.g. for MovieLens data, only ~3 percents of the users in the whole dataset were used in the experiments' graph --any particular constraints in limiting the size of the data?

## Presentation and writing:

- In introduction it was unclear to me why the unknown node distribution lead to the use of MDP -- as the authors noted "the underlying (bipartite) graph instances may come from the same unknown distribution...", for a fixed unknown distribution there could be other simpler estimators. I'd suggest the authors add more motivation and related prior works to explain why the use of RL and MDP became the focused solution

- In section 2.1 Figure 1 was first-time referenced but the plot did not show up until page 5, and jumping from section 2.1 to figure 1 made me confused because non of the MDP notations  / framework was introduced before or in sec 2.1

- I found the arrangement of related work section between the setup section and the MDP model section rather hard to read. Section 2 described the problem setting but it felt that the related work section then distracted the reader.

- as noted above please add missing details in the related works

## Typos:
- Section 2.1  "..the online setting involves reasoning under uncertainly making the design of optimal online algorithms non-trivial .."

---

> ### Author Response · Authors · 2022-08-28
> **Reply to Reviewer 6WVx**
>
>
> Thank you for reviewing our paper! Please let us address your concerns:
>
> ## Regarding the pointed out weaknesses:
>
> ### Prior work:
> We have added a reference to the survey studying relevant algorithms in the KIID setting. Since our framework follows the Unknown i.i.d setting, we found it misleading to the reader if additional detail about the KIID setting was to be added.
>
> ### Baselines:
> To the best of our knowledge, our work is the first applying RL to online matching problems in the unknown i.i.d setting. Other papers mentioned in the related work section apply RL in the offline setting.
>
> We thank the reviewer for pointing out the Q-learning paper. Wang et al., 2019 does focus on a variant of the OBM problem. However, they relax the standard constraint of making an instantaneous and irrevocable decision of matching or skipping. This is done by batching arriving nodes through an RL agent and then applying an offline matching algorithm to the batch. We believe this approach has several critical shortcomings that make it not comparable to our setting. First, the real-time instantaneous matching of an arriving node is, in large part, what makes OBM a hard problem. Second, the proposed solution is specifically tailored for the E-OBM problem and cannot be easily adapted to other variants.
>
> The greedy baselines have been proven to perform well on online problems (see Borodin et al. [1] for a study on OBM). Moreover, in an effort to compare to a strong baseline which takes advantage of past data, we developed greedy-t which uses the best threshold from the training set. For the OSBM problem, we only use the greedy baseline as previous work has shown it to be the strongest in our setting [2].
>
> [1]  ACM J. Exp. Algorithmics, ISSN 1084-6654, doi:10.1145/3379552. URL https://doi.org/10.1145/3379552.
>
> [2] Proceedings of the AAAI Conference on Artificial Intelligence, doi:10.1609/aaai.v33i01.33011877. URL https: //ojs.aaai.org/index.php/AAAI/article/view/4013.
>
> ### IID Assumption:
> The RL framework could be used in other settings. However, we generate training data by sampling nodes from the base graph at random. In order for our dataset to be representative of the actual distribution, we would need the nodes to be IDD. The framework, however, could be used in non-IDD settings if the graph generation process were not to be dependent on the IID setting. We have added an explanation to section 5.1 to clarify this point.
>
> Regarding the adversarial setting, the model can learn the adversarial graph (unlike “pen and paper” algorithms which do not learn to optimize over a specific input). One may wish to study the setting where the adversary is also a learning model (and arrive at a min-max formulation of the problem). As mentioned in the paper, it is mostly the case where the arrivals are IID in practical domains, which is why we chose to focus on the IID setting.
>
> ### RNNs:
> We note that MDPs by design have the Markovian property which means we only need to look at the current state to make an action. Therefore, we do not see how RNNs would be beneficial in this setting. The IID assumption means that the node arrivals are independent from previous arrivals. The sequential input to an RNN, however, would be suited for when node arrivals have a direct impact on each other.
>
> Another main issue with using an RNN would be that it is non-invariant to the order of past node arrivals i.e. hidden state will be different depending on the order. We note that the optimal matching would be the same on a graph regardless of the order at which the nodes arrive.
>
> Additionally, RNNs are often hard to train on long time horizons due to the vanishing/exploding gradient problem.
>
> ### OSBM graphs:
>
> In the case of OSBM, we follow the setup by Dickerson et al. [2] which samples the users with most ratings and uses them as fixed nodes. This ensures that the edge weights calculated based on the average ratings per genre are accurate. Hence why ~3 percent of the users in the whole dataset were used.
>
> ## Regarding Presentation:
>
> ### MDP and Estimators:
> We note that the MDP is used as a natural way to formulate the OBM problem, it is not used as an estimator of any kind. We use different Neural Network models as estimators to learn implicit statistical properties which are hard to learn otherwise. As we note in the paper “Unlike pen-and-paper algorithms, our use of historical information is not limited to estimating the arrival distribution of incoming nodes. Rather, our method takes advantage of additional statistics such as the existing (partial) matching, graph sparsity, the |U|-to-|V | ratio, and the graph structure. Taking in a more comprehensive set of statistics into account allows for fine-grained decision-making.”
>
> ### Fig 1:
> Thanks for pointing this out. We have removed the reference to this figure in section 2.1. The figure is also rightfully referenced in section 4.1.

---

### Review · Reviewer_kv87 · 2022-08-19

**Summary Of Contributions:**

This paper propose a reinforcement learning method to solve the Online Bipartite Matching problem. A set of policy networks are designed to improve the matching efficiency. Extensive experiments on several synthetic and real-world datasets show its significant performance improvement over previous baselines.

**Broader Impact Concerns:**

None.

**Requested Changes:**

Questions:
1.What is the impact of the supervised module on the performance? There is no ablation study about this factor? What is the relationship between the SL model and the RL model?
2.What is the training curve?
3.How to evaluate the RL policy? Do you test the policy in the same environment?
4.The main algorithm is Reinforce. I suggest authors can try some state of art RL algorithms, such as TD3 and SAC.
5.Supervised learning methods such as Deep FM are widely used in recommendation tasks. However, the SL baseline is not compared in this paper.
6.I suggest that this paper clearly discuss the contribution of the performance improvement of each feature type. That is, what is the most important part?



Presentation comments:
1.The definition of the action in page 6, denoted by pi_t as immediate actions confuses the readers. I suggest a_t instead.
2.It would be better to explain 3 feature types: graph-level, incoming node features, and solution-level features.
3.L(pi) in the loss function of Reinforce should be an estimate of the return, not the loss.
4.It is more readable to present results in Figure 4 in a table.



**Strengths And Weaknesses:**

Strengths:
1.This paper study an interesting application of RL to real world problems.
2.The related work is detailed.
3.The proposed network structure is novel and convince readers.
4.The performance improvement is significant.

Weakness:
1.The experimental evaluation of the model is not sufficient.  Some ablation study of each part in the model is missing.
2.The technical contribution of this paper is weak, as there is no special design of the training algorithm in the matching domain.

---

> ### Author Response · Authors · 2022-08-28
> **Reply to Reviewer kv87**
>
> Thank you for taking to the time to review our paper. Please let us address your concerns:
>
> ### Regarding the pointed out weaknesses:
>
> 1. Thank you for your assessment. We have multiple models that explore the role of the RL framework, history, and invariance on the performance. For example, ff-hist outperforms ff and ff-supervised due to its use of history and RL training, respectively. The gnn-hist architectures adds graph-based learning and provides some benefits for OSBM on MovieLens (Figure 8). As such, the 6 architectures in section 4.2 and their experimental evaluations are indeed ablations. We would appreciate it if the reviewer could provide an explanation on what experiments should be added to the paper to make the experimental evaluation sufficient.
>
> 2. The main contribution of the paper is formulating online combinatorial optimization problems under the unknown i.i.d setting as an MDP and designing model architectures based on real world applications of the problems. We note that previous work on RL for combinatorial optimization focuses on offline optimization problems where existing MIP solvers are still stronger and more practical. However, in OBM, existing baseline policies do not perform well across different real-world data distributions. Our RL MDP formulation nicely fits the OBM problem, and allows for the design of custom online policies that are shown to outperform baselines and generalize well to different graph sizes. We believe our framework is novel in that regard.
>
>
> ### Regarding the Questions:
>
> 1. There is no supervised module. We do present a separate supervised model which learns via behavior cloning. The results indicate that the SL model does not generalize well and suffers from overfitting (as noted in our results section). Another downside of supervised learning is training requires labels which is infeasible for NP-hard variants of OBM.
>
> 2. Appendix B includes details about training. We do provide an insight into the convergence during training by providing a sample plot indicating the average validation reward during training (see fig 9- Appendix B.2).
>
> 3. It is common practice to choose a “simpler” learning algorithm, especially in the RL for combinatorial optimization setting (see [1,2] for examples). This is because (i) the contribution of this work (and similar papers) is about the problem formulation, model architecture, etc. (ii) More complex methods such as SAC and TD3 introduce more sensitive hyperparameters that are often hard to tune.  It is thus more effective to have a “simpler” learning algorithm with more reliable results. We have found that Reinforce, when used with an exponential baseline, produces consistently strong results.
>
> 5. The DeepFM architecture is aimed to address recommendation tasks, where the goal is to estimate the probability a user will click on a recommended item. However, the goal in matching problems is to match nodes in a network. The two tasks are different; matching is a combinatorial optimization problem whereas recommendation is a prediction problem. In recommendation tasks, the nodes in the network have a choice and the task is predicting if an edge would form in the network depending on the choice of the nodes. The goal in matching problems is to match nodes in a network (and not to predict if an edge would form in future. The nodes have no choice in matching problems).
> On a related note, we show in our paper that supervised models are not fit for OBM. Mainly because they require labels and cannot adapt to different scenarios like RL (ie. the well known advantages of behavior cloning vs RL).
>
> 6. Each feature type is meant to help the model understand the state of the current graph and matching to better reason about the future. For example, graph-level features give the model an estimate of weight distribution for each fixed node U. This can help to make a more informed decision on whether to skip or match. Our results in Figure 5 demonstrate this.
>
> ### Presentation comments:
>
> - Thanks for pointing out to the loss function. $L(\pi)$ is the negative of the return (there is no estimate of the return since the returns are deterministic given an action). We have clarified this in the paper.
>
> - Please refer to table 1 for an explanation on feature types.
>
> [1] Wouter Kool, Herke van Hoof, and Max Welling. Attention, learn to solve routing problems! In International Conference on Learning Representations, 2019. URL https://openreview.net/forum?id= ByxBFsRqYm.
>
> [2] Hanjun Dai, Elias B. Khalil, Yuyu Zhang, Bistra Dilkina, and Le Song. Learning combinatorial optimization algorithms over graphs. In I. Guyon, U. V. Luxburg, S. Bengio, H. Wallach, R. Fergus, S. Vishwanathan, and R. Garnett (eds.), Advances in Neural Information Processing Systems, volume 30. Curran Associates, Inc., 2017. URL https://proceedings.neurips.cc/paper/2017/file/ d9896106ca98d3d05b8cbdf4fd8b13a1-Paper.pdf.

---

### Decision · Action_Editors · 2022-09-23

**Recommendation:** Accept as is

**Comment:**

All three reviewers agree that this paper deserved an accept. The primary weakness across reviewers is that the novelty side of the work is not so high. That said, as one reviewer points out, the authors study a natural approach that it is useful for the community to understand and see performance on. Moreover, the authors studied their approach quite thoroughly. The reviewers also felt that the author responses successfully addressed the questions of the reviewers.